# Application of Metaverse Service to Healthcare Industry: A Strategic Perspective

**DOI:** 10.3390/ijerph192013038

**Published:** 2022-10-11

**Authors:** Chang Won Lee

**Affiliations:** Healthcare MBA Track and School of Business, Hanyang University, Seoul 04763, Korea; leecw@hanyang.ac.kr; Tel.: +82-2-2220-2790

**Keywords:** metaverse civilization, metaverse service, strategic perspective, healthcare industry

## Abstract

This study is to explore a state of the art in metaverse service that is an emerging issue in applying it to the healthcare industry. The purpose of this study is to provide applicable strategic scenarios for effective metaverse service planning and implementation in healthcare settings. This study is focused on metaverse service as a business model. Thus, related literatures of metaverse service are reviewed in various aspects in healthcare industry. An exploratory approach is used to analyze current qualitative data characterizing healthcare metaverse service business positions and derive applicable strategies from business trends of current metaverse services. Several cases are examined based on the data obtained from various sources of healthcare and other related industries. This study synthesizes finding results and suggests applicable strategies of metaverse service in the healthcare industry. This study will facilitate strategic decision-making and policy-making processes to pursue a business opportunity development through an application of a metaverse service in healthcare and similar settings.

## 1. Introduction

With the 4th industrial revolution and the COVID-19 pandemic, a huge and new form of digital civilization is emerging. In a modern civilization, the most important entity that connects the individual and society is a corporation. In the new digital civilization, the metaverse is the topic of technological innovations powering business opportunities. The metaverse related business environment has changed dramatically due to the COVID-19 situation. The existing consumer culture that centered on face-to-face business is converted to zero-contact business culture, and online business is more activated in all types of business industries. Among these industries, the healthcare industry using smart technologies became very common and pervasive. As the scope of smart service applications has been expanded and the usage of smart service has exploded, clearly the zero- contact business culture has had a great influence on the healthcare industry.

Applications of metaverse services have been initiated and has become an increasingly promising and significant area of the healthcare industry [1,2,3,4,5]. Therefore, it is important that the healthcare industry be aware of the values and commitment to metaverse services as an essential aspect of strategic planning for the long-term vision in the healthcare industry. The opportunities on metaverse services in the healthcare industry have potential but are complex and conflicting since the opportunities are different in each sub-industry. It is very difficult to properly balance current metaverse service requirements of a business model and its relevant technology without a systematic approach to evaluate potential business opportunity development. If it is overlooked in the planning and evaluation of the business opportunity development in metaverse services in the healthcare industry, the metaverse service strategy may fail to fulfill the compelling demands of the market and satisfying various customers unmet needs. If a certain goal in metaverse service is chosen with little consideration for other goals, the strategic decision can be costly both in terms of technology and operations plannings concerns.

One of the important activities is the strategic planning for business opportunity development of metaverse services in the healthcare industry. The existence of various key factors makes strategic planning in business opportunity development more difficult to plan and implement. It is more complicated since many qualitative and quantitative factors must be incorporated in the strategic planning process of a business opportunity development in a metaverse service in the healthcare industry. Various studies related to metaverse in the healthcare area have been explored [6,7,8,9,10]. However, few previous studies conducted to identify applicable aspects of metaverse service in terms of a strategic perspective in the healthcare industry.

The purpose of this study is to explore applications for business opportunity development using metaverse services in the healthcare industry in terms of a strategic perspective. This study will aid in developing the healthcare industry’s metaverse service business pertinent to strategic planning. The relevant literatures and cases of metaverse are reviewed based on the data acquired from various sources in the healthcare industry and other industries. Thus, this study is to identify insights and strategies for effective planning and implementing based on the importance of metaverse as a new healthcare service. Specifically, this study is (1) to explore a metaverse timeline and driving forces, along with what is metaverse, how it has developed, and what is its structure through existing literature research and data, (2) to present case studies on metaverse-related business corporations in a healthcare industry, and (3) to provide strategic findings for directions of the metaverse adoption and implementation in the healthcare industry. This study will strengthen the healthcare industry’s ongoing strategies to meet necessary market and technology requirements while positioning the healthcare industry to respond to new business opportunity development in metaverse service.

## 2. Metaverse Service Background

### 2.1. Conceptual Timelines

The term metaverse first appeared in the science fiction novel Snow Crash by Stephenson. It is about an immersive virtual realm accessible through virtual reality (VR) goggles, depicting a virtual reality world in which people use their digital avatars to explore the online world to escape from often dystopian reality. Since then, with the rapid progress and growth of internet technology services, various innovative technology services have been provided to allow service users to experience numerous new services with more virtual interactions in cyberspace. In the novel, the metaverse was described as a virtual world that could be accessed using audiovisual output devices such as goggles and earphones, and economic and social activities were possible. In 2003, a service called Second Life was launched and became popular and received attention [11,12,13,14,15].

Metaverse can be defined as a 3D transcendent world that converges the physical, digital, and biological worlds using a hyper-connected, intelligent new technology that impacts every economy and industry. Metaverse service is an integrated service between physical, augmented, and virtual reality. Users in the metaverse service can express themselves using their digital identities. If the basic infrastructure is provided to individual users, it provides the necessary service environment for all users to have a converged experience of the virtual and real worlds. Metaverse can be widely used in the sense of a living-type and game-type virtual world where both real and unreal coexist in the overall aspect of politics, economy, society, and culture through digital identities such as avatars. With the development of technologies such as artificial intelligence (AI), virtual reality (VR), augmented reality (AR), Internet of Things (IoT), Blockchian, Non-Fungible Token (NFT), and other technologies implementing metaverse services, the transition to the metaverse civilization that provides a new level of experience by combining virtuality and reality is accelerating [16,17,18,19,20,21].

Four essential technologies in the metaverse service are virtual reality (VR), augmented reality (AR), mixed reality (MR), and extended reality (XR). VR is a technology service that enables users to experience a real life-like environment in a virtual world created by digital appliances. AR is a technology service that provides an environment that interacts with a virtual object represented in 2D or 3D with a real space. MR is a technology service that combines real world and virtual world information to create a virtual space where the two worlds are combined. XR is a technology service that implements a concept that encompasses VR, AR, and MR as well as another form of reality that will appear in the future. XR is a general-purpose technology that implements the metaverse and is considered as a key means of convergence with major industries such as the healthcare industry to build a new industrial ecosystem. As the demand for non-face-to-face technology services continues due to COVID-19, the XR market will grow even more rapidly in the future. As the era of XR conversion is expected to approach, the industry will need to be prepared to develop business opportunities [22,23,24,25,26,27].

Table 1 shows the metaverse timeline with its signposts. Until now, virtual environments of numerous services and applications from the social world to the virtual game world have been developed through immersive experiences and digital innovation. However, most of them are not integrated into the platform and exist in a stand-alone form, failing to show the standard and consistency of the system. In this context, the metaverse implies the cyber physical world beyond the simple compounding of the two words meta and universe. Current metaverse civilization trend is creating a new world of shared virtual services, powered by advanced technologies such as 5G and 6G networks, virtual reality, and artificial intelligence. Among these technologies, artificial intelligence technology improves the immersive experience of virtual agents and shows the importance and possibility of big data processing to realize human-like intelligence.

The American non-profit organization Accelerated Studies Foundation (ASF) presented an alternative concept in the metaverse roadmap announced in 2007 [47]. ASF moved away from the dichotomous approach that viewed the metaverse as an alternative or opposite to the real world, and suggested the metaverse as a junction, nexus or convergence between the real world and the virtual world. This conceptual evolution follows that elements of the real world such as objects, devices, actors, interfaces, and networks are inevitably accompanied in the implementation and use of the virtual environment. Although the metaverse and the virtual world are regarded as the same concept in some practical cases, it is a recent trend to view the phenomenon as the complex of the real world and the imaginary world due to the continued influence of the metaverse roadmap. In this point of view, an immersive technology is a linkage between a real world and imaginary world so that these two worlds become one world called the metaverse, a complex world [48,49,50,51,52].

Table 2 presents characteristics among four different metaverse types to compare in terms of concepts, features, values to implement, core technologies in each type, and some challenges or side effects from each category [47]. In general, metaverse reminds a lot of virtual reality, but it is a broad topic that includes concepts combining reality and technology, such as augmented reality, lifelogging, the mirror world, and virtual world. It records and shares daily life through wearable devices. Although AR is less immersive than VR, it has the characteristic of being highly likely to be used in everyday life. The AR concept, which first appeared in the late 1990s, is implemented through mobile phones, computers, and mechanical devices when a virtual object is overlaid, and a smart environment can be built using GPS information and networks. The mobile game “Pokemon Go” which was released in 2017 and gained worldwide popularity is one of the examples of the AR. The concept of life logging allows you to store information about your life and share it with other users by storing it on a server, it existed before the 21st century, but with the spread of mobile phones, it is most often used in daily life with the mirror world. Social media such as Facebook, Instagram, Tweet, and Kakao Story are included in life logging type. Most of the map-based services fall into the mirror world. Google Earth, which collects satellite and street photos from around the world and periodically updates them to reflect the ever-changing appearance of the real world, and offline accommodation by copying an individual’s home into a virtual space. Airbnb that is used for reservations is a representative example of the mirror world.

### 2.2. Metaverse Driving Forces

There are several driving forces on metaverse’s re-emergence that has been gradually developed as ASF announced the Metaverse roadmap, but it has been expanded with the release of the beta version of Omnibus, an open 3D design collaboration platform, by Nvidia in 2020. Nvidia CEO Jensen Huang recently used the phrase as an idiom, using the expression that the Omnibus beta release, “metaverse is coming”. It is analyzed that the re-emergence of the metaverse is due to the complex action of socio-cultural, economic, and technological factors that have occurred in the last 5 to 10 years. A non-face-to-face society has become a new normal due to the COVID-19 pandemic, the 5G and 6G communication network, and large amounts of information processing at high speed. The recent metaverse boom creates a new digital civilization led by the MZ generation those who are the mid to late 1990s as starting birth years and the early 2010s as ending birth years. As they stay at home for a long time due to the COVID-19, metaverse has become the ultimate enabler as a communication channel for daily activities [48,53,54,55].

In a social aspect, a game researcher and designer Jane McGonigal argues that the growing demand for games stems from a desire to find a way to a broken reality. When GameBeat introduces a planning panel with the theme of “Into the 2020 Metaverse,” it mentions that the metaverse is attracting attention as a haven for reality. The metaverse can function as a space for imagining and creating a better world rather than an escape from reality.

In a cultural aspect, with the spread of digital media, users have grown from passive consumers to actively enjoying and creating content. Many metaverse services, including the sandbox-type virtual world, provides open creation tools that allow users to materialize and reproduce imagination and creativity. Cultural transformation that uses the metaverse service for the purpose of forming and maintaining social relationships rather than for the purpose of consuming content are increasing due to the COVID-19 pandemic.

In a technological aspect, a device-platform-network ecosystem has been advanced. With the Samsung Gear VR in 2015, Facebook Oculus and HTC Vive in 2016, VR devices have commercialized and advanced in the late 2010s. Since the mid-2010s, AR/VR content development platforms such as Unity, Amazon Sumerian, and Google Poly toolkit have rapidly increased. With the commercialization of 5G in 2019, a network environment has been prepared to reduce latency problems. Gartner predicts that the demand for immersive displays will surge after reaching the stage of popularization of related technologies.

In an economic aspect, Accenture predicts that XR technology in 2019 has grown rapidly through 2023. As consumption expenditure is expected to increase at least 2 to 3 times in B2B and B2C over the next three years, a trend of vitalization of national and private investment is expected [56,57,58,59,60].

## 3. Application of Metaverse Service to Healthcare Industry

An exploratory methodology is used to analysis qualitative data obtained from various sources and literatures. The synthesized findings are derived so that necessary strategies for metaverse service strategies are provided. Among metaverse technologies, XR- and VR-based technology for metaverse services are promising. Thus, XR-based metaverse products and services are explored. In 2025, the global AR and VR market will be around 815 billion dollars and the global immersive technology related healthcare market is expected to reach $5.1 billion [61,62]. The healthcare industry is the field where XR is expected to be most actively used. XR technology is being used in a variety of ways from the education and training of healthcare professionals to practices of surgery and patient care. XR is mainly used for the purpose of supporting medical training and rehabilitation treatment. XR virtualizes the patient’s condition to determine the exact surgical location for treatment.

Johns Hopkins University succeeds in the world’s first surgery using Xvision, an AR-based spinal surgery support system. It has obtained FDA approval and is being used in actual surgery from 2020. In patient treatment, VR is used for rehabilitation, mental illness, and psychotherapy to motivate patients to continuously participate in training, maximizing the therapeutic effect. As an application case of AR, it is possible to develop a system for smooth operation by medical staff and improve the efficiency of patient health management. It can be applied to information and medical history inquiry about a specific patient, and to develop an operation assistance system. In the healthcare education field, through VR simulation, tasks such as diagnosis, treatment, and surgery are virtually trained in the same situation as the actual field. In the operating room, AR technology can be used to directly monitors additional surgical information on the affected area to improve surgical concentration and accuracy [63].

In the future, with the development of hand gesture input technology, if complicated medical procedures can be implemented and voice control is able to activate, XR technology is actively used in telemedicine and surgery other than healthcare education. As an example of innovation in the healthcare industry through XR, it is possible to maximize the treatment effect by not only reducing training costs, improving surgical accuracy and safety, but also reproducing a specific healthcare environment and providing a high sense of immersion beyond the limitations of the real treatment environment.

Four representative cases for XR development in healthcare industry are explored and provided their corporate identification, product/service areas, and their descriptions [64]. Tetra Signum developed a non-face-to-face medical education platform service that improved the virtual education platform service with contents for surgical education. It also develops VR CPR Training Solution. Recently, in a situation where exchanges between foreign personnel are limited, a university hospital uses the platform of Tetra Signum to conduct remote lecture discussions and observation of remote operating rooms with doctors from 8 countries including Japan, Singapore, and the UK.

Surgical Mind provides a medical training service using human body models and VR technology instead of real patients. Aesthetic plastic surgery training service and cataract surgery training service are developed for effective surgical training of doctors and medical students. Aesthetic plastic surgery training service is a service for cosmetic surgery and procedure training developed with a virtual engine. It can be implemented to learn how and where to inject with detailed anatomical information of the face. It can learn where the syringe tip is in the skin precisely within the 0.5 mm error range.

Techno Village’s Rehaveware is a fully immersive VR rehabilitation treatment service for recovering lost movement functions of patients with known brain diseases such as stroke, Parkinson’s disease, and brain surgery among others. This service is effective in treating hemiplegic disorders in patients in highly immersive virtual reality. Linking with the Internet of Things (IoT) such as smart globes and smart balls is expected that the patient’s desire to exercise increases and the therapeutic effect improves. In the future, as well as brain disease rehabilitation treatment, it is planned to expand and apply to psychological treatment products for potential dementia patients as well as children and adolescents suffering from family violence or serious diseases.

DHU’s Senior Wealth Center develops cognitive rehabilitation contents and physical rehabilitation contents for brain injury patients using VR devices and song gesture recognition devices to improve cognitive function of high-risk groups of mild cognitive impairment and dementia. Virtual reality cognitive rehabilitation content program VR-AIN is a full 3D graphic realization service. It is designed to provide patients with a sense of immersion and realism so that rehabilitation training and treatment can be performed just like in a real environment with five main cognitive domains like perception, memory, attention, passing, and execution.

XR-related metaverse has shown achievements in the fields of treatment such as patient surgery and development of medical personnel. A representative case is the AR-based spinal surgery support system X-Vision of Augmedics. X-vision helps for the doctor to see the patient’s spine structure realized in AR overlapping the surgical site so that the surgeon can get help with the exact surgical location and procedure. X-Vision has received an approval from the U.S. Food and Drug Administration. In 2020, the first X-vision spine surgery successfully completed. SentiAR’s holographic cardiac ablation guidance service, CommandEP, is to provide visualization of patient anatomy information required during cardiac surgery with a mixed reality (MR). It is to use as a medical imaging service that enables the review, analysis, communication, and exchange of multifaceted digital images.

Table 3 shows application examples of metaverse related technologies in healthcare industry. AR-based glasses can apply AR technology to support healthcare professionals by enabling hands-free documentation during a wound treatment. A new AR system called HoloLens developed by Microsoft uses for collaboration scenarios like minimal-invasive surgery. It is an interaction model to support collaboration in a certain care space with other users. Legacy AR uses for collaboration among multiple head-mounted displays (HMD) users. This case presents collaboration among a single user and others who join the space by sharing the view of the HMD user. The co-workers participate remotely through Skype-enabling tablets or PC’s. The World Health Organization (WHO) utilizes AR for healthcare professionals to learn life-saving skill such as personal protective equipment (PPE) and cardiopulmonary resuscitation (CPR). It also educates COVID-19 responders and healthcare workers how to use virtual reality to treat people with mental and emotional difficulties. Stryker, a US medical technology company, improves the surgical operating room design process by using Microsoft’s AR smart glasses, since 2017 [65,66].

Table 4 presents a meta-analysis of metaverse service application literatures in healthcare industry. Fifteen application areas are identified along with their description and their authors. Most studies reviewed are conducted within recent one or two years. Studies related to bibliometric review and care related activities and decision studies are most common studies. Other various cases in healthcare are explored in aging care, cardiovascular cancer, chronic diseases, dentistry, fetal and gynecology, emergency among others.

## 4. Metaverse Service Strategy

It is very important from the perspective of a company’s future strategy whether the metaverse is established as a service industry ecosystem in a digital civilization. This new business ecosystem achieves balanced regional development and job creation by developing an entrepreneurial economy that enhances social and global wellness. For this, if the industry-academia-government cooperates, a virtuous synergetic progression will result in. Establishing a future strategy for digital innovation by metaverse is very important, and the effort to successful implementation is the life of the country and the future world. It is directly related to the leap of a new civilization. Thus, systematic analysis and efforts will be required in terms of a strategic perspective. To achieve sustainable growth in the future based on the innovation of the existing technology model, it is necessary to promote the proper strategies and policies. It may also consider in accordance with the internal and external environment of the metaverse industry and changes in the strategic direction at local and global concern.

Under these circumstances, two major environmental problems are emerging. First, the dynamics of the global metaverse market and demand are rapidly increasing. This means that changes in the metaverse business environment are accelerating. Specifically, global infrastructure is emerging continuously one after another and the need to strengthen industrialization for an immersive economy is emerging. The immersive economy is an economy and civilization that creates new industrial, social, and cultural values that converge reality and virtuality while expanding the realm of experience in terms of time and space. The competition of metaverse products as a service industry is expected to intensify, the high-cost structure and growth restrictions appearing in the early industry are expected due to a shortage of development manpower, and the social capacity to promote the metaverse service industry is still lacking. To solve these issues, future industry research that reflects global trends should be conducted effectively, and research and efforts to build a metaverse system are required. The second realistic situation is the problem of strengthening the capabilities of the metaverse industry in response to changes in local and global environments. Locally, industrial structure, service diversification, new business development, and eco-friendly policy issues should be established. Globally, the expansion of metaverse technology development, the establishment of infrastructure and platform facilities, and securing competitiveness in the new economy are essential considerations.

This new civilization requires a virtual platform business that operates in a virtual world different from the existing platform. This is a huge disruptive innovation. As with the early days of business, it is not easy to see tangible results. As the user’s usage culture changes, the individual and social ability to absorb new technologies must be supported. There is not yet a large technology company in many countries that pursue such aggressive and disruptive innovation. Most current metaverse businesses are implementing the metaverse service on an existing business platform. The reason why artificial intelligence (AI) is indispensable for the next-generation metaverse industry is that unlike the existing metaverse service that limits immersive experiences due to insufficient data, AI allows users to freely create creative content with huge new users and behaviors. This is because it provides a rich infrastructure to create data sources as well as deploy them within the platform.

Following specific strategies and relevant tasks in each strategy can be utilized in a similar setting. Each strategy and task are in general and is easily applicable to a healthcare industry (see Table 5) [86,87].

From an individual perspective, as various jobs such as game developers, virtual costume designers, and virtual architects related to the healthcare industry can be created within the metaverse platform, individuals in the metaverse civilization can discover and utilize opportunities in newly emerging jobs, startups, and secondary life. Creators such as bloggers and YouTubers in the 2D web era are evolving into creators in the 3D metaverse era. Just as YouTube created a new job as a YouTuber and became a place of opportunity for individuals, innovation created by individuals will continue the metaverse platform.

From a corporate perspective, hospitals and healthcare corporates need to find ways to innovate in productivity and discover cooperative business models. Currently, several global corporates are using the metaverse business platform. It is necessary to set a proper strategy to use metaverse to innovate the way to work. Corporates should find ways to innovate productivity using metaverse environments for all industries and value chains, and productivity innovation can be achieved through designing metaverse experiences for each industry. Healthcare corporates can innovate the metaverse experience by utilizing digital humans and various IP collaborations with the metaverse platform. It will be necessary to review the use of digital human in various aspects such as users/patients service and public relations.

From a policy perspective, the policymakers and government have established and are proceeding with future strategic plans to respond to the metaverse era. For example, it is establishing strategies that can apply to healthcare industry. In the field of healthcare education, future-oriented healthcare education methods such as metaverse educational centers/clinics that have evolved from existing offline centers/clinics. Healthcare administrative and legal fields are also areas to pay attention to. Through the evolution of the existing 2D-based e-administrative or legal services to 3D-based virtual services, patients and stakeholders can receive more efficiently healthcare administrative and legal service in the virtual world. Legal work is to provide civil complaints and legal services using digital avatars. Another consideration can be given to introducing and utilizing metaverse for medical training and qualification tests suitable for the non-face-to-face era. In the current method of using simulated patients and mannequins, it is possible to consider application plans in detailed and practical fields such as healthcare performance evaluation and clinical techniques using digital twins.

Virtual convergence economy development strategy can be initiated as a strategy to respond to the metaverse civilization. The virtual convergence economy refers to an economy that uses metaverse to expand economic activity spaces such as work, leisure, and communication from reality to virtual convergence spaces to create new experiences and economic values. It is expected that metaverse is applicable to use in all stages of the value chain of a healthcare industry to accelerate innovation in traditional business models and drive economic growth. Virtual convergence technology is an interface that connects the real world and the digital world. It promotes the coexistence of the real and the virtual and overcomes the physical limitations of reality. The virtual convergence economy is rapidly emerging as an economy that creates new added value in the overall economy as the use of metaverse is greatly expanded due to the rapid growth of non-face-to-face environments due to the development of metaverse technology and COVID-19 and the digitalization of the industrial base.

## 5. Conclusions

This study is to explore business applications of metaverse service in the healthcare industry by using an exploratory meta-analysis method. This study is focused on identifying business strategies of metaverse service in a healthcare industry. Relevant data was obtained from various literatures and available online sources. Analysis was performed and synthesized findings followed by providing business strategies applicable to the healthcare industry. The metaverse civilization is fast approaching. Nevertheless, around half of global population is still unable to utilize the internet civilization properly and digital risks are a reality [88]. As time goes by, various metaverse related service and platforms appear and attract attention in the healthcare industry. Additionally, with the advent of the COVID-19 experiencing users, who are familiar with the virtual world, interest in various healthcare activities to be performed through the metaverse service is increasing. MZ generation accustomed to creating a new self in the virtual world through SNS since childhood creates and expresses themselves through a new self in the digital world, that is, multi-persona. Through new innovations and reforms, companies should focus on increasing the value of customers and users so that everyday life can be conducted more safely and healthily. For the metaverse service to have universal value as a new civilization in a healthcare industry, it will need to take root in an open innovation ecosystem for further R&D and business expansion. As non-face-to-face care service has become common due to COVID-19, it has become comfortable to communicate and care in a virtual environment.

The literary origin of the metaverse is William Gibson’s VR cyberspace called The Matrix in the 1984 science fiction novel Neuromancer [89]. An experience economy has been arrived in various service areas as economic systems [90]. The term metaphysics in a modern philosophy is used extensively to include questions about the reality of the external world, the existence of other minds, the possibility of a priori knowledge, and the nature of sensation, memory, abstraction, etc. In mathematics, there are three numbers such as a real number, imaginary number, and complex number. Likewise, convergence of a real world (physical world) and imaginary world (cyber world) will give a complex (metaverse) world. Just as the real world and the virtual world are different, a civilization that is completely different from the current civilization will appear in the metaverse world. This world view in business civilization will give an unprecedented business opportunity in a healthcare industry. Thus, metaverse service in a healthcare industry will be an ultimate co-value creation between a provider and a user individually and collectively. Studies for technological advancement have been conducted through neural networks, computer vision, and other pertinent techniques to improve the quality of the computational results and to advance conceptual frameworks [14,21,47,79,86,91,92].

This study has study limitations such as study time horizon, study scope, and study subjects to cover. Thus, many study areas exists to explore for a future study. Other metaverse service technologies unmentioned in this study may provide another opportunity where patients and their families and patients and doctors can freely communicate. It also builds more complex metaverse service environment in conjunction with artificial intelligence (AI) technologies by providing patient-tailored immersive healthcare services. One of important areas to explore is about medical twin and other fundamental metaverse technology to create a virtual space identical to reality to predict the treatment effect and to prescribe the optimal drug. Medical twin is a concept that applies digital twin service technology to the healthcare field that creates identical twins in virtual space and verifies them through various tests. Healthcare metaverse services and its related technologies in this study are not the only examples of metaverse service technology being used in the healthcare field. Additional study may be conducted focused on individual business strategies (micro-level) and business ecosystem strategies (macro-level) in terms of profit model development and market analyses.

## Figures and Tables

**Table 1 ijerph-19-13038-t001:** Metaverse timeline with its related product and service.

Year	Descriptions	References
1992	Stephenson coined metaverse term and concept in a novel Snow Crash	[11]
1993	Gelernter proposed the digital twin concept	[28]
2003	Linden Labs launched internet-based virtual world Second Life	[29]
2004	Unity released a platform for creating and operating 3D web media	[30]
2006	Roblox released an online game platform and game creation system	[31]
2008	Nakamoto invented bitcoin	[32]
2011	Blockchain established as the first bitcoin blockchain explorer	[33]
2014	Facebook acquired VR tech company Oculus	[34,35]
2015	Microsoft launched a head mounted display (HMD) AR device HoloLens	[36,37]
2016	Niantic launched a popular mobile AR game Pokemon Go	[38,39]
2017	Epic released an online video game Fortnite	[40,41]
2019	Facebook announced social VR platform Horizon	[42]
2020	Google acquired AR glasses start-up North (Thalmic Labs)	[43]
2020	LG Uplus released the world’s first 5G-based AR glasses	[44]
2020	NVIDIA released a real time graphics collaboration platform Omniverse	[45]
2021	Microsoft released digital collaboration platform MS Mesh	[46]

**Table 2 ijerph-19-13038-t002:** Characteristics of four different metaverse service types.

Item	Augmented Reality	Life-Logging	Mirror Worlds	Virtual World
Concept	An interactive environment by superimposing virtual 2D or 3D objects in real space	Technology to capture, store, and share everyday experiences and information about things and people	A virtual world that reflects the real world as it is, but expanded informationally	A virtual world built with digital data
Features	Building a smart environment using location-based technology and networks	Recording information about objects and people using augmented technology	Utilizing virtual map, modeling, GPS, and lifelogging technology	Interacting between avatars reflecting the user’s ego
Value/Needs	Provides immersive content that combines the real world, fantasy, and convenience	Extensive real-world experience and information can be checked at any time and shared with others	Maximized usability by integrating and expanding external information into the virtual space	Provides a new virtual space that does not exist in reality where various individuals can perform activities
Core Technology	Unstructured data processing, 3D printing, 5G networks	Online platform, ubiquitous sensor, 5G network	Blockchain technology, GIS system, data storage and 3D technology	Graphics technology, 5G network, artificial intelligence, block chain technology
Challenges	Confusion in augmented reality space where reality is superimposed Ownership of characters in augmented reality, etc.	Infringement of portrait and property rights and leak of internal secrets Violation of dual position ban, etc.	Unfair trade occurred due to information manipulation problem and huge platform lock-in effect	Avoidance of the real world and fear of disorder that will cause moral and ethical problems

**Table 3 ijerph-19-13038-t003:** Metaverse Application in Healthcare Industry.

Areas	Description	Benchmarking Companies
Surgical Operations	Providing navigation solutions to surgeons during surgery It can penetrate the patient’s anatomy and perform surgery with high precision and minimal complications It can also be used for remote virtual guidance and support	Proximie [67], Augmedics [68], Vicarious Surgical [69]
Healthcare Education	Providing a visual, hands-on learning experience for beginners and experienced healthcare professionals to improve their understanding of patient anatomy and how complex surgical operations are performed	Osso VR [70], Medivis [71,72], FVRVS (FundamentalVR) [73,74]
Treatment	Providing new treatments for patients suffering from chronic diseases The use of AR/VR devices helps in relaxation and stabilization of patients suffering from chronic pain. Used to provide safe and controlled virtual exposure to traumatic events Accelerate patient cognitive rehabilitation and recovery through AR/VR technology	AppliedVR [75,76], Oxford VR [77], XRHealth [78,79]

**Table 4 ijerph-19-13038-t004:** Meta-analysis of metaverse service literature review in healthcare industry.

Application Area	Description	Authors
Aging care	Providing social connection to the elderly	Cho [7]
Cardiovascular health	Applications of the metaverse on cardiovascular health	Mesko [80], Skalidis et al. [10]
Chronic diseases	Managing chronic diseases	Sun et al. [26], Southworth et al. [81]
Care related activities and decisions	Applying Blockchain, NFTs, IoT and other related technologies	Bhattacharya et al. [23], Thomason [24], Mejia & Rawat [25], Musamih [57], Yang et al. [58], Mozumder et al. [21]
Conceptualizing and bibliometric review	Characterizing and reviewing metaverse in healthcare	Yang & Lee [12], Damar [15], Garavand & Aslani [16], Yang et al. [18], Chen & Zhang [20], Sun et al. [26], Song & Chung [59], Ghanbarzadeh et al. [3],
Dentistry	Discussing the metaverse in dentistry	Afrashtehfar & Abu-Fanas [19], Duman et al. [82]
Emergency	Applications of the metaverse on emergency care	Wu & Ho [83]
Gynecology	Applications of the metaverse on gynecology and fetal health	Werner et al. [84]
Healtrhcare environment	Exploring new economic opportunities and socioemotional environments	Thomason [5], Yu et al. [27], Mejia & Rawat [25], Lee [54]
Medical education and training	Tooling for information sharing, clinical simulation, healthcare delivery	Holloway [1], Lee et al. [9], Wiederhold & Riva [13],
Mental health	Applications of the metaverse on mental health	Almarzouqi et al. [22], Usmani et al. [53], Turbyne [55]
Oncology	Applications of the metaverse on cancer care	McWilliam & Scarfe [8], Zeng et al. [85]
Ophthalmology	Opportunities and challenges in eye care	Tan et al. [52]
Oral health promotion	Providing virtually to target groups and communities through Metaverse software	Albujeer & Khoshnevisan [6]
Wound management	Designing a wound management application	Klinker et al. [60]

**Table 5 ijerph-19-13038-t005:** Strategies and tasks for the metaverse service development in healthcare industry.

Strategy	Tasks
Strategy on healthcare education and training	Reviewing the introduction of Metaverse applications for healthcare training, education, and qualification tests suitable for the non-face-to-face era Looking for detailed application methods such as clinical performance evaluation (CPX) and Objective Structured Clinical Examination (OSCE) using meta-human in the method using the current simulated patient and mannequin (dummy) Enhancing healthcare training and education using metaverse application Nurturing high-quality talents with master’s and doctoral degrees of metaverse discipline in healthcare industry
Strategy on full utilization metaverse technology to resolve various healthcare issues	Innovating healthcare industry by conversing technological advantages in different industries Creating an investment base for metaverse utilization throughout the region for reducing digital healthcare divide Laying the foundation for expansion, such as the metaverse fund, which will drive private investment participation in healthcare industry Preparing to respond ethical issues resulting from metaverse service in healthcare sector
Strategy on advancing healthcare business environment	Supporting for healthcare device core technology and finished product development Constructing all-round metaverse data dam of clinical and not-clinical big data in healthcare industry Expanding ultra-high-speed, minimum-delay metaverse service through advanced network development to apply to healthcare immersive experience With the development of new healthcare business models suitable for metaverse environments
Strategy on support for securing global competitiveness in healthcare industry	Nurturing healthcare unicorn business in 2030 with intensive support Securing future innovative technologies such as metaverse technology visualization and five senses technology applicable for healthcare industry Promoting of metaverse corporate globalization Preparing metaverse promotion law and major regulations improvement in preparation for borderless among hospitals and customers, doctors and patients, and countries

## Data Availability

Not applicable.

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
