# Peer review of "Application of Metaverse Service to Healthcare Industry: A Strategic Perspective"

_ijerph, 2022, doi:10.3390/ijerph192013038_

Round 1
Reviewer 1 Report
I value the opportunity to review your manuscript - ijerph- 1918746
The following are my comments:
(1) The author should provide a literature review in a table form of brief findings on the application of metaverse in healthcare industry as the authors presented numerous references on the said topic.
(2) Table 2 needs some references
(3) The word MZ generation is not common connotation in other countries, please explain more.
(4) Many stat numbers indicated by the authors don’t have references.
(5) I was surprised by the abrupt change of subject. After providing sample of application examples of metaverse related technologies in healthcare industry. The author just mentioned “This study investigates the use of audio-based device and physical interaction with the AR based glasses in a within-subjects design experiment.” I am not sure why there is a random choice of technology that will be investigated. And this one was only explained in brief one paragraph.
(6) The last paragraph of (3. Application of Metaverse Service to Healthcare Industry) was not clear – the author keeps giving a lot of examples which are not based on the cases provided. The author needs to clarify clearly how many cases provided and provide clear connections of the cases with the topic because based on how the author wrote the cases is just like randomly giving examples.
(7) Regarding the 4. Metaverse Service Strategy, the challenges and strategy given by the author are too general. The author should provide more specific challenges and strategy for healthcare industry.
(8) It is not clear why the author explained 7 types of metaverse platform and from an individual perspective? Can the author make clear connections? I think the author can remove this part and focus more on healthcare industry and the corporates.
Based on all of this, I think your manuscript is not publishable in its current format. Therefore, I am recommending you make minor revisions of your work. I hope you find my comments helpful.
Author Response
Reply to Reviewer 1’s Comments
Thank you for your valuable comments. Here are point by point replies to comments. Reflection is made as much as possible in the current study scope.
(1) The author should provide a literature review in a table form of brief findings on the application of metaverse in healthcare industry as the authors presented numerous references on the said topic.
- Requested table form is made in Table 4, p. 8.
(2) Table 2 needs some references
- Necessary references are added in p. 4.
(3) The word MZ generation is not common connotation in other countries, please explain more.
- Required explanation is provided in p. 5.
(4) Many stat numbers indicated by the authors don’t have references.
- Additional corrections are made. Unnecessary numbers and stats are removed in p. 6.
(5) I was surprised by the abrupt change of subject. After providing sample of application examples of metaverse related technologies in healthcare industry. The author just mentioned “This study investigates the use of audio-based device and physical interaction with the AR based glasses in a within-subjects design experiment.” I am not sure why there is a random choice of technology that will be investigated. And this one was only explained in brief one paragraph.
- Corrections and made. Unnecessary sentences are removed.
(6) The last paragraph of (3. Application of Metaverse Service to Healthcare Industry) was not clear – the author keeps giving a lot of examples which are not based on the cases provided. The author needs to clarify clearly how many cases provided and provide clear connections of the cases with the topic because based on how the author wrote the cases is just like randomly giving examples.
- Corrections and necessary explanations are made. Unnecessary sentences are removed.
(7) Regarding the 4. Metaverse Service Strategy, the challenges and strategy given by the author are too general. The author should provide more specific challenges and strategy for healthcare industry.
- More specific challenges and strategies are provided focused on healthcare industry in pp. 9-10.
(8) It is not clear why the author explained 7 types of metaverse platform and from an individual perspective? Can the author make clear connections? I think the author can remove this part and focus more on healthcare industry and the corporates.
- Necessary revision is made. Related parts are removed.

Reviewer 2 Report
This paper offers an overview of metaverse services in the context of the healthcare industry. As it stands, the paper lacks clarity and focus. Please see below the aspects that need consideration.
§ As it stands, the paper doesn’t tell the reader whether it’s a review paper or merely an opinion. Please clarify this aspect upfront in the abstract section as to what this paper is about, if it is a review article then you need to increase coverage and relevant references. if this is not a review article then what sort of research is being conducted in this article? What scientific bases it has? what models were adopted, devised and applied to come up with conclusions? – It is not clear to the reader because it looks more like a commentary and opinions rather than tangible, quantifiable, research.
§ Author has briefly mentioned the timeline of the metaverse in Table 2, pls. provide corresponding references here.
§ Similarly, the metaverse service strategy section is weak with not much retrospective comparison with the state-of-the-art. Please improve coverage here.
§ As the functioning of a metaverse requires a combination of several cutting-edge technologies like Virtual Reality, Augmented Reality, Artificial Intelligence, machine learning, Internet of Things (IoT), and so on, this aspect is completely ignored in this paper which needs to be addressed, in particular, emerging technologies and so on.
Please include the following articles which describe some of these aspects.
§ Area efficient architecture for large-scale implementation of biologically plausible spiking neural networks on reconfigurable hardware - DOI: 10.1109/FPL.2006.311352
§ Overview: Technology Roadmap of the Future Trend of Metaverse based on IoT, Blockchain, AI Technique, and Medical Domain Metaverse Activity - DOI: 10.23919/ICACT53585.2022.9728808
Accelerated Diagnosis of Novel Coronavirus (COVID-19)—Computer Vision with Convolutional Neural Networks (CNNs) - https://doi.org/10.3390/electronics11071148
Author Response
Reply to Reviewer 2’s Comments
Thank you for your valuable comments. Here are point by point replies to comments. Reflection is made as much as possible in the current study scope.
As it stands, the paper doesn’t tell the reader whether it’s a review paper or merely an opinion. Please clarify this aspect upfront in the abstract section as to what this paper is about, if it is a review article then you need to increase coverage and relevant references. if this is not a review article then what sort of research is being conducted in this article? What scientific bases it has? what models were adopted, devised and applied to come up with conclusions? – It is not clear to the reader because it looks more like a commentary and opinions rather than tangible, quantifiable, research.
- This study is more qualitative study focusing the state of the art on the metaverse service.
- Revision is made to reflect reviewer’s comments in the abstract and the main body.
- Author has briefly mentioned the timeline of the metaverse in Table 2, pls. provide corresponding references here.
- Necessary references are added p. 3.
- Similarly, the metaverse service strategy section is weak with not much retrospective comparison with the state-of-the-art. Please improve coverage here.
- Necessary improvement is made. Improvement is made focused on healthcare industry in p. 10.
- As the functioning of a metaverse requires a combination of several cutting-edge technologies like Virtual Reality, Augmented Reality, Artificial Intelligence, machine learning, Internet of Things (IoT), and so on, this aspect is completely ignored in this paper which needs to be addressed, in particular, emerging technologies and so on.
- Other key technologies are added in the related section. Since this study is focused on a business service and strategy perspective, all technological concerns are not mentioned. For future study, some strategic analyses can be conducted with technological forecasting of future metaverse service to expand and vitalize future business market.
Please include the following articles which describe some of these aspects.
- Three papers are properly reflected accordingly.

Round 2
Reviewer 2 Report
Thanks for addressing my comments. I am happy to accept the paper.